# Temporal deposition of copper and zinc in the sediments of metal removal constructed wetlands

**Zeinah Elhaj Baddar** [ORCID] *, **Erin Peck, Xiaoyu Xu**

Savannah River Ecology Laboratory, University of Georgia, Aiken, South Carolina, United States of America

* Zeinah.Baddar@uga.edu

## Abstract

The objective of this study was to explore the effects of time, seasons, and total carbon (TC) on Copper (Cu) and Zinc (Zn) deposition in the surface sediments. This study was performed at the H-02 constructed wetland on the Savannah River Site (Aiken, SC, USA). Covering both warm (April-September) and cool (October-March) seasons, several sediment cores were collected twice a year from the H-02 constructed wetland cells from 2007 to 2013. Total concentrations of Cu and Zn were measured in the sediments. Concentrations of Cu and Zn (mean ± standard deviation) in the surface sediments over 7 years of operation increased from $6.0 \pm 2.8$ and $14.6 \pm 4.5$ mg kg$^{-1}$ to $139.6 \pm 87.7$ and $279.3 \pm 202.9$ mg kg$^{-1}$ dry weight, respectively. The linear regression model explained the behavior and the variability of Cu deposition in the sediments. On the other hand, using the generalized least squares extension with the linear regression model allowed for unequal variance and thus produced a model that explained the variance properly, and as a result, was more successful in explaining the pattern of Zn deposition. Total carbon significantly affected both Cu ($p = 0.047$) and Zn ($p < 0.001$). Time effect on Cu deposition was statistically significant ($p = 0.013$), whereas Zn was significantly affected by the season ($p = 0.009$).

## 1. Introduction

Constructed wetlands are green, man-made alternatives to treat urban, industrial, and agricultural runoff, storm water, and municipal and industrial wastewater [1–3]. Compared to wastewater treatment facilities, and owing to their low construction and operational costs, constructed wetlands are relatively cheaper and require less maintenance [4–6]. The design of constructed wetlands simulates natural wetlands and provides similar but more controlled ecosystem functions [7,8]. These functions include the biogeochemical cycling of carbon and nutrients and removal of contaminants such as heavy metals [9–11].

Heavy metals could enter constructed wetlands through point sources such as industrial wastewater [12], and nonpoint sources such as agricultural runoff of products incorporating heavy metals (e.g., micronutrient fertilizers, pesticides, etc) [13]. The removal of heavy metals from wastewater in the free surface wetlands is often achieved through several processes such

**Funding:** This research was supported by the National Nuclear Security Administration-Tritium

and the U. S. Department of Energy, through Financial Assistance Award No. DE-FC09-96SR18546 to the University of Georgia Research Foundation. The funders had no role in study design, data collection and analysis, decision to publish, or preparation of the manuscript. Disclaimer— This report was prepared as an account of work sponsored by an agency of the United States Government. Neither the United States Government nor any agency thereof, nor any of their employees, makes any warranty, express or implied, or assumes any legal liability or responsibility for the accuracy, completeness, or usefulness of any information, apparatus, product, or process disclosed, or represents that its use would not infringe privately owned rights. Reference herein to any specific commercial product, process, or service by trade name, trademark, manufacturer, or otherwise does not necessarily constitute or imply its endorsement, recommendation, or favoring by the United States Government or any agency thereof. The views and opinions of authors expressed herein do not necessarily state or reflect those of the United States Government or any agency thereof.

**Competing interests:** The authors have declared that no competing interests exist.

as sedimentation, filtration, adsorption, and uptake by macrophytes and other organisms [14]. Thus, heavy metal enrichment in wetland soils is expected.

Wetland sediments are considered as sinks to contaminants such as metals [15,16]. While the transport of heavy metals from the wastewater to the underlying sediments is highly desirable prior to discharge into natural water bodies, the deposition of heavy metals in the wetland sediments should be closely monitored overtime, to ensure that it will not become toxic to benthic organisms and microorganisms, where the trophic transfer could likely happen [17–19]. Also, as wetlands mature, their efficiency in contaminant and nutrient removal may decline [20,21].

Heavy metals, especially Cu and Zn, are strongly associated with organic matter [22–24]. Wetlands tend to have a higher deposition rate of organic matter compared to other ecosystems, due to the high rate of carbon fixation through photosynthesis, coupled with the slow decomposition rates due to the dominant anaerobic conditions [9]. Thus, wetland sediments with higher organic matter content will likely become enriched with metals over time, which occurs through several processes such as complexation and sorption to humic substances [25].

Redox potential and pH have major effects on the biogeochemical cycling of heavy metals. In wetland systems, two distinct redox zones could be identified; an aerobic zone close to the surface and an anaerobic zone that lies deeper in the sediments [9]. The availability of organic matter in the anaerobic zones enhances the biotic reduction of iron and sulfates, which results in the precipitation of Cu and Zn into metal sulfides and/or their adsorption onto iron hydroxides. Under acidic conditions, most metals will be outcompeted by the abundant $H^+$ ions for adsorption onto negatively charged surfaces such as clay minerals and organic matter. On the other hand, neutral to alkaline sediments in the wetland systems result in metal fixation and reduce the toxicity to macrophytes through adsorption onto negatively charged surfaces [3].

The efficiency of a constructed wetland in the removal of contaminants is usually assessed by measuring and comparing the concentrations of these contaminants in the inflow and the outflow effluent. Few studies investigated metal retention in constructed wetlands and sediments over time and seasons [25–27]. The design of the constructed wetlands plays a major role in determining the sustainability of the system in contaminant removal [28]. Hydraulic parameters, macrophytes and substrates used to provide a continuous supply of organic matter and nutrients important for plant growth, could also affect the performance and efficiency of a wetland system [29].

The H-02 wetland was constructed in 2007 to treat storm runoff and industrial effluent generated from the Tritium facility on the Savannah River Site (Aiken, SC, USA). The wetland follows the traditional free surface flow design and aims at the removal of heavy metals from the effluent prior to discharge to the Upper Three Runs Creek which eventually reaches the Savannah River. The objective of this study was to report and evaluate the potential temporal, seasonal, and TC effects on the deposition of Cu and Zn in the H-02 wetland sediments over the course of 7 years (from 2007 to 2013). We used linear models (lm) and the linear model with a generalized least squares (gls) extension to resolve residual heterogeneity and help understand the deposition behavior of Cu and Zn in the H-02 wetland.

## 2. Methods

### 2.1. Study site

A schematic diagram of the H-02 wetland system showed the sediment core sampling sites in both wetland cells 1 and 2 (S1 Fig). A detailed description of the free-surface wetland system can be found in Xu and Mills, 2018 [11] and Xu, et al 2019 [30]. Briefly, the Tritium facility discharges wastewater through source pipes into a rectangular pool where water is retained to

provide hydrological control. The wastewater then goes through a splitter box that roughly splits the flow equally into two adjacent rectangular wetland cells (WC1 and WC2) of about 2240 m$^2$ area each. The wetlands are lined with geo-synthetic impermeable material, covered with 46–61 cm of native wetland soil that has been amended with gypsum (source of sulfur (S)), fertilizer, and organic matter to provide a substrate that would support the growth of plants and microbial communities [11]. The cells were planted with the macrophyte, giant bulrush (*Schoenoplectus californicus)*, which also contributes to immobilizing metals in wetland sediments by providing organic ligands that act as binding sites, while providing a carbon source for sulfate reducing bacteria, which in turn, help sequestering Cu and Zn from the water column through the formation of mineral ZnS and CuS [31]. Water stays for 48 hours in each wetland cell before leaving at the other end through a pipe which discharges water into the Upper Three Runs Creek before eventually ending into the Savannah River.

## 2.2. Sediment core collection and processing

Sediment cores were collected from each wetland cell for metal analysis twice a year, once during the warm seasons (April-September), and once during the cool seasons (October-March) starting from June 2007. Each wetland cell was divided into 3 transects. The focus was on the first and last transects as they represent the locations at which water enters (inflow) and leaves (outflow) each wetland cell. The cores were put on ice, transferred to the lab, and stored upright at 4°C until the next day to allow the flocculent layer to settle. Surface standing water was carefully siphoned off from the cores avoiding disturbing the flocculent layer. The siphoned cores were stored in the freezer for at least a week prior to further processing. Then each core was extruded into several sections based on changes in color and texture. In most cases, cores had three distinct layers; a top layer which contains the upper sediment layer mixed with flocculent material that is rich in organic matter, a middle clay-like textured layer, and a bottom sandy-textured layer. Extruded core sections were transferred to 50 mL metal free tubes and freeze dried (Labconco Corporation; Kansas City, MO, USA) until constant weight. Dried sediments were passed through a 2 mm sieve.

## 2.3. Chemical analysis

US EPA methods 3051A [32] and 6020A [33] were both used to process the sediments and analyze total metal concentrations, respectively. Briefly, microwave acid digestion (CEM MARSxpress, Matthews, NC, USA) in 10 mL concentrated HNO$_3$ at 180°C for 10 minutes was performed on freeze-dried sediments, and ICP-OES (Optima 4300 DV, PerkinElmer, Waltham, MA, USA) was used to measure the total concentrations of Cu and Zn in diluted digests. Standard reference material, digestion blanks, duplicates, and spikes were used as measures of QA/QC throughout the analytical process. Method detection limits for Cu and Zn were, respectively, 2.05 and 2.06 ng g$^{-1}$. Marine sediment standard reference material MESS-3 (National Research Council of Canada ((NRC - CNRC); Ottawa, Canada) had an average percent recovery of 92.2 and 93.0% for Cu and Zn, respectively (n = 2). Relative percent differences among replicates were 13.2% for Cu and 11.7% for Zn (n = 4). Average spike percent recoveries for Cu and Zn were 116.6% and 112.2%, respectively (n = 3). Total carbon (TC) and total nitrogen (TN) were measured in the freeze-dried sediments using the US EPA method 9060A [34].

## 2.4. Statistical analysis

**2.4.1. Data processing.** Dates of sediment core sampling were split into two main categories; "warm" or "cool" based on the temperatures in Aiken (SC, USA). Months from April to

September and from October to March fell into the warm and cool seasons, respectively. We split the full data set into several lists based on wetland cell (1 or 2), transect (inflow or outflow), and sediment core depth (top, middle, and bottom). For non-available data, we replaced these values with the average of that variable in the corresponding list. In instances where below detection limit data were encountered, we replaced those with half the detection limit.

**2.4.2. Data exploration.** Metals, TC and TN concentrations were log-transformed to the base 10 to minimize the effect of potential outliers on the analysis [35]. Data exploration included constructing several conditioning plots that showed the relationship between the explanatory and the response variables. Metals (Cu and Zn) and TN and TC as well, were far more abundant in the top layer in the sediment cores compared to the lower layers which had similar and much lower TC and metal content. Thus, the major focus of this analysis was on the surface (top) layer of the sediment-named surface sediments hereafter. Previous work performed in the H-02 wetland has also shown that the surface layer had significantly higher Cu and Zn concentrations compared to the other two lower layers which were not significantly different from one another [36].

Also, since the outflow location is more critical for monitoring metal concentrations as the effluent leaves the wetland into the regulatory Upper Three Runs Creek, our focus was on samples collected from the outflow location in both wetland cells. Since TC and TN are proxies of organic matter, which is highly responsible for metal complexation, we initially included both parameters as fixed effects in model construction. Data analysis and modeling were performed using R-studio 4.0.2 [37].

**2.4.3. Model selection.** Model selection was performed following the processed mentioned in [38]. The process is described in detail elsewhere (S1 Text, S2 Fig). Briefly, after eliminating fixed variables of high correlation (S1 Table, S3 Fig, S1 Text), the linear model (lm) was tested for residual homogeneity. In case of residual heterogeneity, a variance covariance structure was used, and once the homogeneity of residuals was achieved, the backward selection method was applied to include the significant fixed effects and their interactions. Model validation involved testing residual normality, homogeneity, and independence.

## 3. Results

### 3.1. Deposition of Cu and Zn in the surface sediments

Average total Cu and Zn concentrations measured in the surface sediments increased between the years 2007 and 2013 (Table 1). Results are reported as (mean ± standard deviation). Total Cu and Zn concentrations in the surface sediments were respectively 6.0 ± 2.8 and 14.6 ± 4.5 mg kg$^{-1}$ dry weight at the beginning of the H-02 wetland operation. After 7 years, total Cu and

**Table 1. Concentration of Cu and Zn in the surface sediment averaged by year, data presented as mean concentration ± standard deviation and (minimum—maximum) concentrations (mg kg$^{-1}$ dry sediment).**

| Year | Total Cu (mg kg$^{-1}$) | Total Zn (mg kg$^{-1}$) |
|------|------------------------|------------------------|
| 2007 | 6.0 ± 2.8 (4.0–8.0) | 14.6 ± 4.5 (11.4–17.7) |
| 2008 | 18.5 ± 13.2 (6.0–32.2) | 34.3 ± 18.9 (14.0–51.4) |
| 2009 | 119.9 ± 139.9 (4.4–349.9) | 177.2 ± 186.2 (7.8–4635) |
| 2010 | 50.2 ± 28.2 (11.7–85.3) | 90.6 ± 52.0 (23.6–170.3) |
| 2011 | 43.7 ± 23.4 (19.8–93.1) | 62.6 ± 33.9 (4.0–106.2) |
| 2012 | 129.7 ± 61.1 (27.0–195.9) | 204.4 ± 96.0 (43.6–296.2) |
| 2013 | 139.6 ± 87.7 (33.5–274.2) | 279.3 ± 202.9 (60.5–674.4) |

For year 2007, n = 2, 2008, n = 3, 2009, n = 5, and for years 2010–2013, n = 8.

**Table 2. Model parameters for Cu and Zn.**

| Metal | Model | Random term/variance covariate structure | Fixed term ($p$ at $\alpha = 0.05$) | Coefficient | $p$-value |
|---|---|---|---|---|---|
| Cu | lm | Cu | Intercept | 226.5 | 0.013 |
| | | | Year | 0.11 | 0.013 |
| | | | $Log_{10}TC$ | 258.30 | 0.047 |
| | | | Year: $Log_{10}TC$ | -0.13 | 0.048 |
| Zn | gls | VarIdent (Year) | Intercept | 1.51 | <0.001 |
| | | VarIdent (Season) | Season/Cool | -0.17 | 0.0093 |
| | | VarFixed ($Log_{10}TC$) | $Log_{10}TC$ | 1.04 | <0.001 |

lm: Linear model, gls: Linear model with generalized least squares extension.

Zn concentrations increased by almost 23.2 and 19.2 times to reach 139.6 ± 87.7 and 279.3 ± 202.9 mg kg$^{-1}$ dry weight of Cu and Zn, respectively (Table 1).

### 3.2. Modeling Cu deposition in the surface sediments

The optimum model for Cu deposition in surface sediments was the linear regression model (lm) (Table 2). This model included neither random effects nor variance covariate terms. Backward selection process showed that the optimum fixed structure included; year, $log_{10}TC$ and a year by $log_{10}TC$ interaction term (Table 2). Model validation showed that the normalized residuals were both homogenous (S4 Fig, Bartlett's test for homogeneity: $p = 0.8729$) and normally distributed (S5 Fig, Shapiro-Wilk test for normality: $p = 0.9358$). The seasonal effects were not statistically significant ($p = 0.9099$) and thus were removed from the final model. Also, Pearson normalized residuals plotted against all main effects (season, year, cell, TC and TN) showed homogeneous distribution (S6 Fig).

Regardless of the year, Cu was positively and significantly ($p = 0.047$) correlated with $log_{10}TC$ (Table 2, Fig 1). There is a significant ($p = 0.013$) and progressively increasing trend in Cu deposition in the surface sediments throughout the years of the study (Fig 2). However, the deposition of Cu did not show significant differences between the warm and the cool seasons ($p = 0.88$, Fig 2).

The interaction between the year and $log_{10}TC$ on $log_{10}Cu$ was statistically significant ($p = 0.048$) and varied across the years (Table 2, Fig 3). Indeed, year to year variation in $log_{10}TC$ deposition resulted in different slopes as depicted in Fig 3. For example, the year 2007 had a slope of 0.83 and much lower $log_{10}Cu$ values compared to the following year where the slope was 2.6 greater than in the former year. This could be attributed to the significant increase in TC deposition in 2008 (after one year operation). In later years, the slope tended to be lower (1.25, 1.29, and 1.13, for the years 2009, 2010, and 2013, respectively). The slope was at it's lowest in the years 2011 and 2012 with 0.60 and 0.64, respectively (Fig 3).

### 3.3. Modeling Zn deposition in the surface sediments

The optimum model for Zn was the linear model with a generalized least square model (gls) extension, which allowed introducing a variance covariance structure for the main effects; season, year, and $log_{10}TC$ (Table 2), in order to describe the heterogeneity in the residuals. The varident structure was used for both season and year while the varfixed structure was used in the case of $log_{10}TC$.

Model validation showed that the normalized residuals were both homogeneous (S7 Fig, Bartlett's test for homogeneity: $p = 0.90$) and normally distributed (S8 Fig, Shapiro-Wilk test

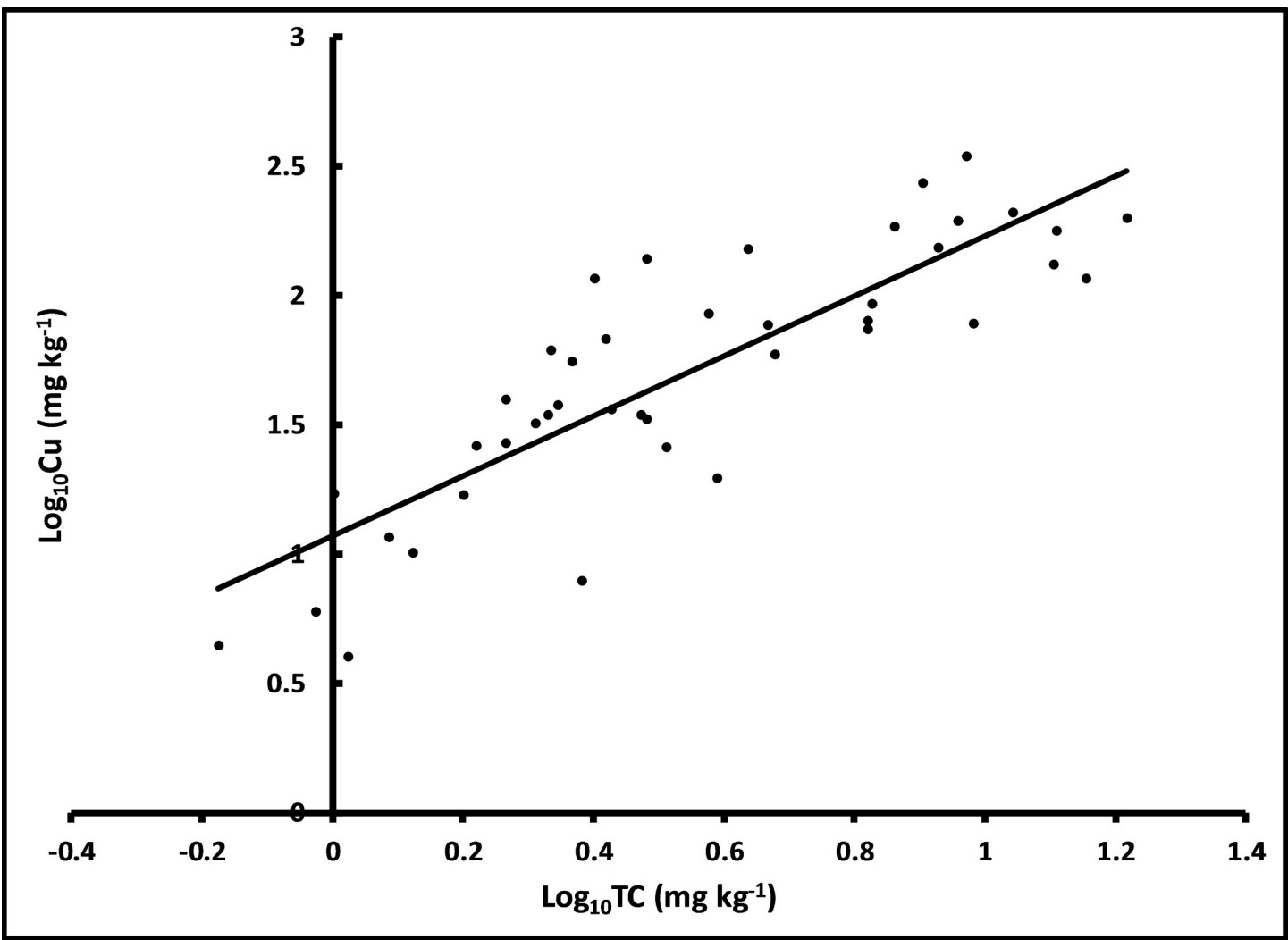

**Fig 1. Relationship between log₁₀TC and log₁₀Cu.**

for normality: $p = 0.87$). The plot of normalized residuals against the main effects did not show any significant patterns, which confirmed residual independence (S9 and S10 Figs).

The effect of total carbon ($\text{Log}_{10}\text{TC}$) on the deposition of Zn in the surface sediment was positive and statistically significant ($p < 0.001$, Table 2) regardless of the year (Fig 4).

There was a significant seasonal effect on the deposition of Zn in the surface sediment ($p = 0.0093$, Table 2). The effect of the cool season was 0.17 times less than the effect of the warm season on Zn deposition (Table 2, Fig 5)

### 3.4. Deposition of Cu and Zn in the lower sediment layers

Although the major focus of this study was on the surface sediments, we reported the concentrations of Cu and Zn in the middle and bottom layers as mean ± 95% confidence intervals (S2 and S3 Tables). Whether averaged by season, location, or by wetland cell, metal concentrations over the years in the middle layer seemed to consistently range between 5.2 ± 5.8 and 19.7 ± 43.0 mg kg⁻¹ for Cu, and between 9.5 ± 4.1 and 28.2 ± 54.1 mg kg⁻¹ for Zn. Likewise, metal concentrations in the bottom layer ranged between 4.4 ± 1.6 and 7.4 ± 7.5 mg kg⁻¹ for Cu, and between 5.6 ± 3.9 and 14.0 ± 7.6 mg kg⁻¹ for Zn.

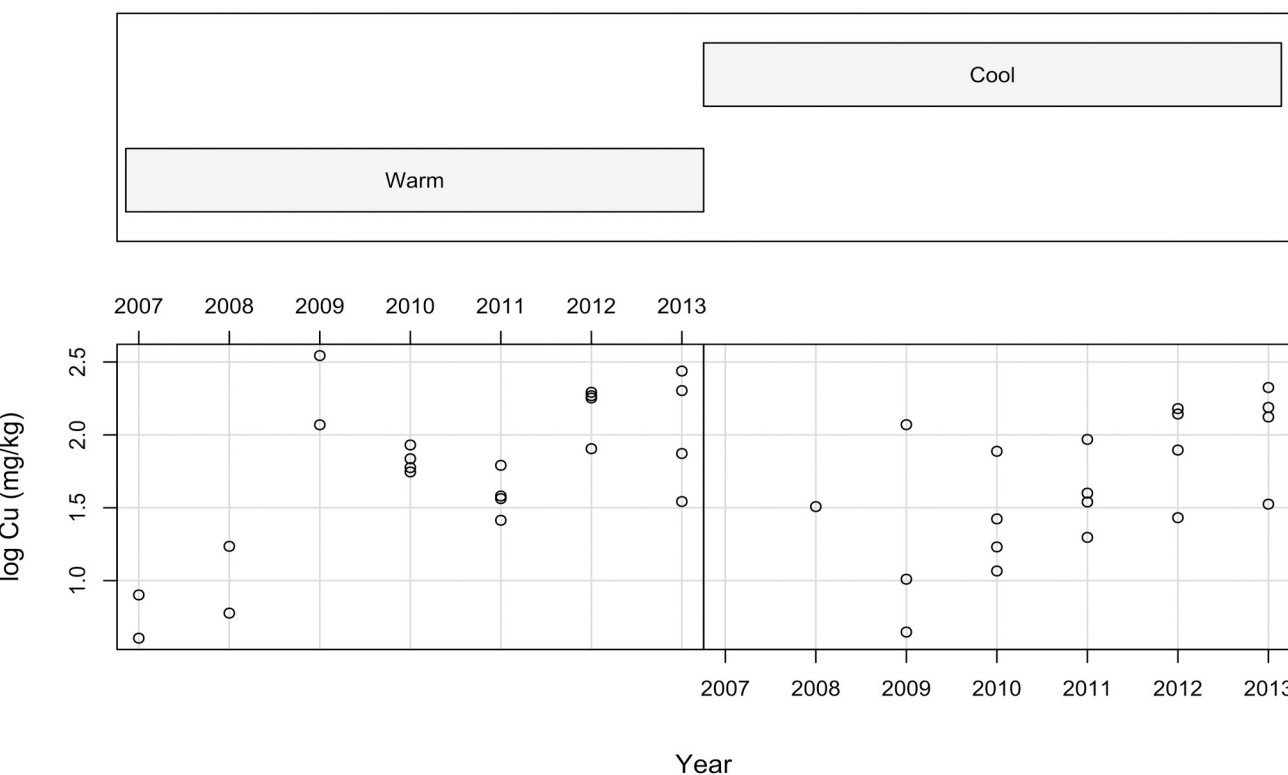

**Fig 2. Conditioning plot showing the pattern of $\log_{10}$Cu deposition in surface sediments (mg kg$^{-1}$) throughout the years 2007–2013 in the warm vs the cool seasons.**

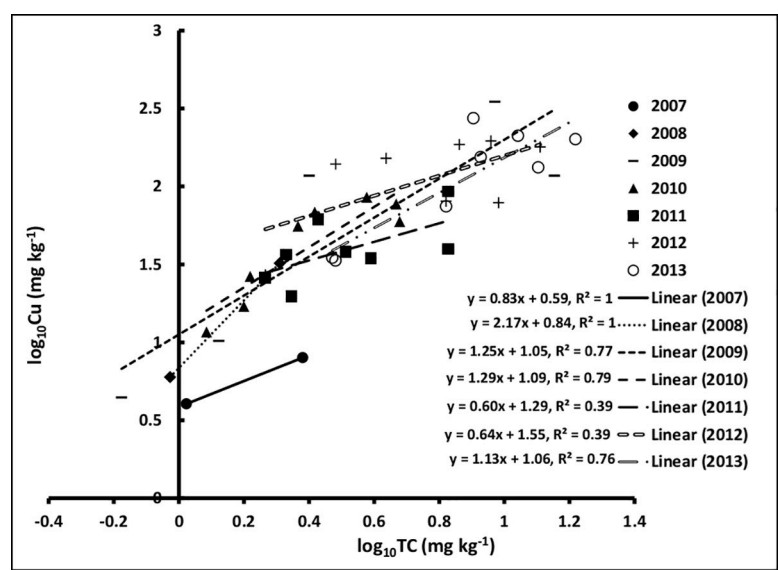

**Fig 3. The effect of the interaction between year and $\log_{10}$TC on $\log_{10}$Cu.**

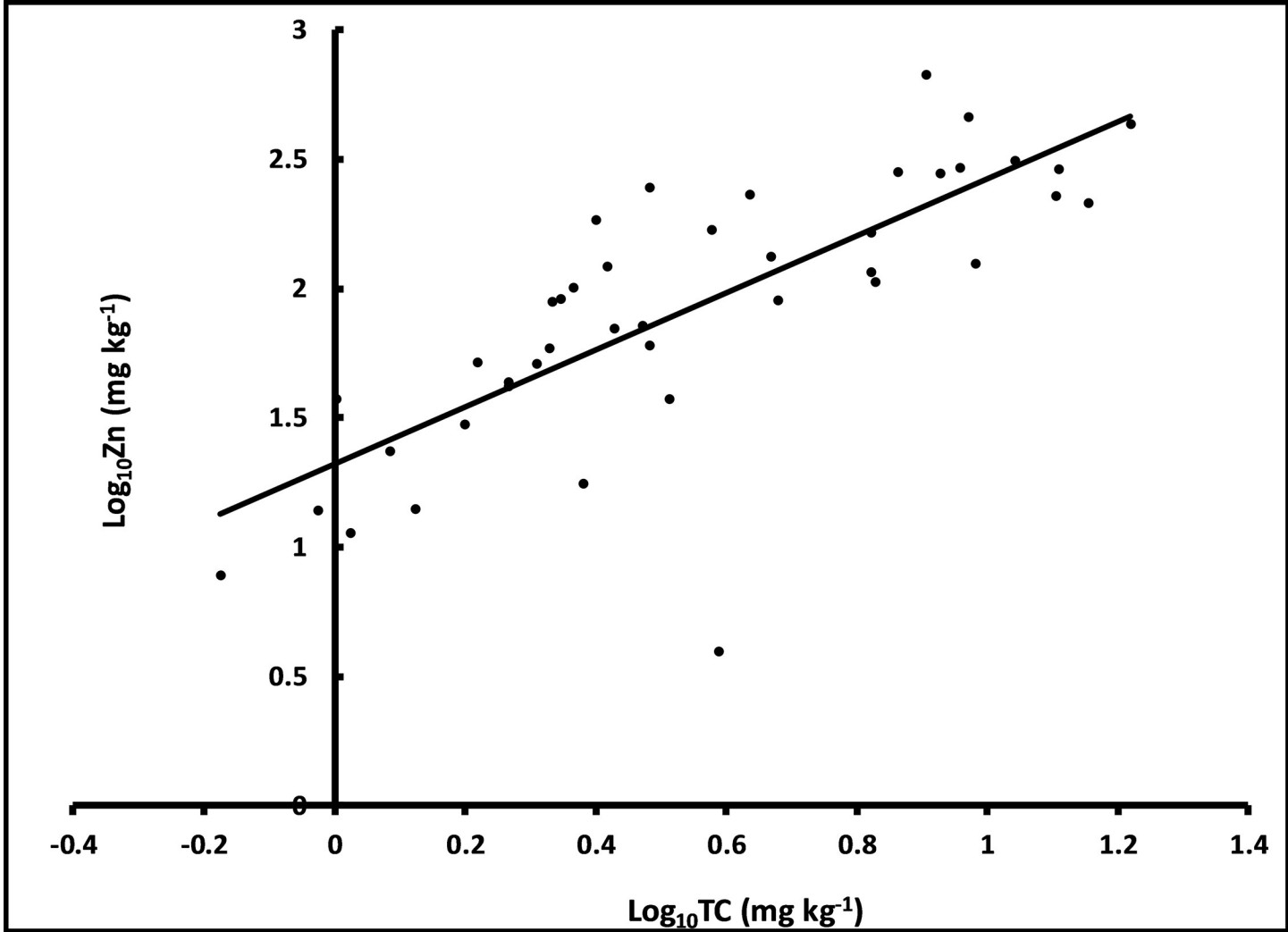

**Fig 4. Relationship between $\log_{10}TC$ and $\log_{10}Zn$.**

## 4. Discussion

This study explored the potential effects of time (7 years), seasons, and organic carbon on Cu and Zn deposition behavior in the surface sediment in the H-02 constructed wetland. The ongoing sampling and analysis of sediment cores are essential to monitor the performance of constructed wetlands over years and seasons. Indeed, this work is a part of several ongoing studies that focus on evaluating the efficiency of the H-02 constructed wetland system in the removal of metals [11,30,36].

### 4.1. Actual concentrations of Cu and Zn in the sediments over the years

The consistent increase in Cu and Zn in the surface sediments over the years implies that the wetland is still successful in the removal of Cu and Zn from the effluent water. Previous work, conducted between the years 2007 and 2018, showed that the H-02 wetland system efficiency in Cu and Zn removal from the Trituim Facility effluent was, respectively, 63.8% and 70.5% [11].

Given : fSeason

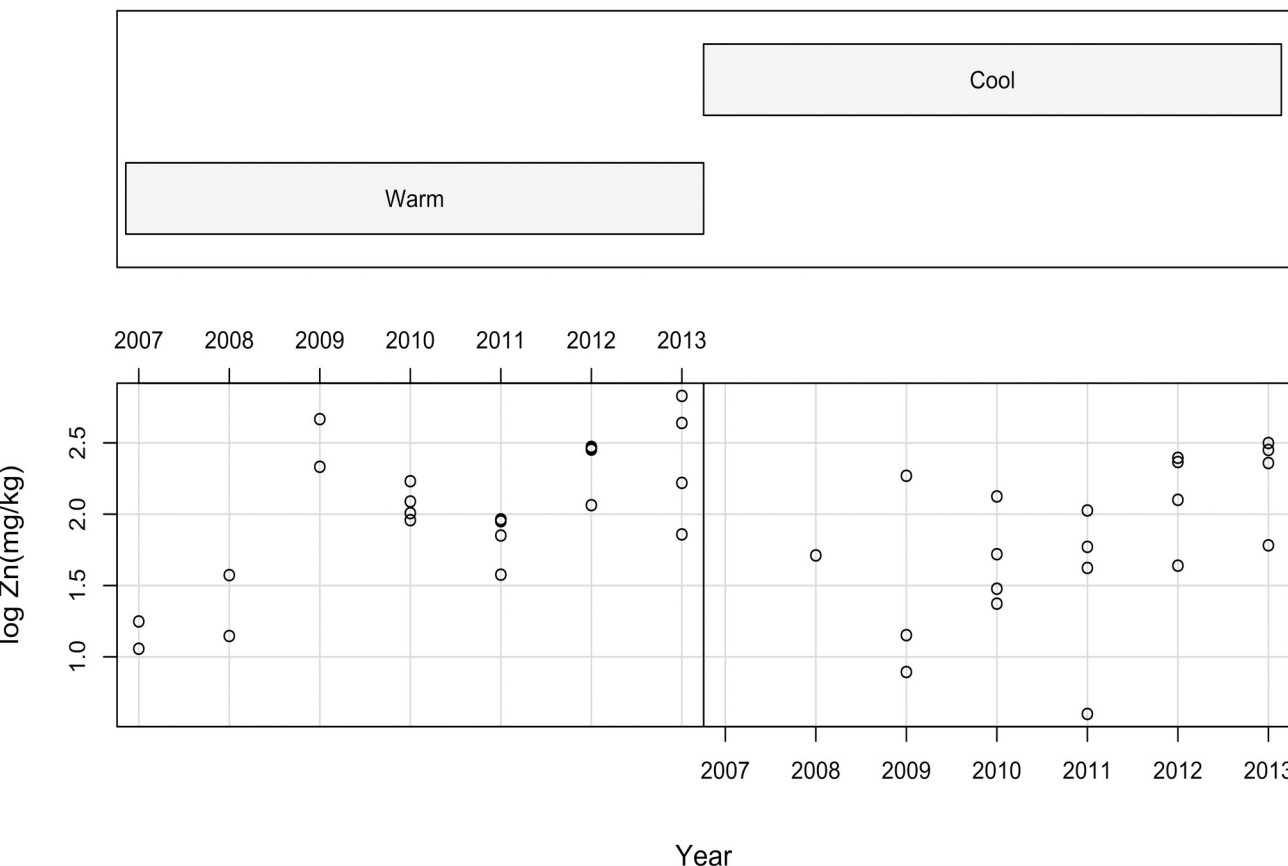

**Fig 5. Conditioning plot showing the pattern of $log_{10}$Zn deposition in surface sediments (mg kg$^{-1}$) throughout the years 2007–2013 in the warm vs the cool seasons.**

According to our previous work [30], the H-02 wetland system is hypothesized to perform in a 3-stage pattern in terms of metal removal efficiency; a steady metal removal performance (plateau stage), a decline in metal removal efficiency (trough stage) and another plateau stage were the metal removal efficiency increases again [30]. In the present study, the deposition patterns of Cu and Zn in the sediments started off by imitating the first plateau stage of operation. Continuous monitoring of metal concentrations in the sediments and water is essential to monitor the efficiency of the H-02 wetland system and to support the abovementioned hypothesis.

### 4.2. Deposition of Cu and Zn in the sediments

Although the data from different seasons, wetland cells, or locations were variable, it clearly showed that Cu and Zn deposition increased in the surface sediments throughout the years. On the other hand, metal deposition in the lower sediments-middle and bottom layers- was much lower compared to the surface sediments and seemed not to change over the years. This is expected due to the design of the H-02 wetland system which operates as a free surface flow system, which implies that the majority of metals removed from the water column would thereby be retained in the top sediment, which is rich in organic matter [36]. Similarly, metal

concentrations significantly decreased with depth in a free water surface constructed wetland that treated industrial effluent [27].

### 4.3. Effect of time, seasons, and carbon on Cu and Zn deposition in the surface sediments

Wetlands are among the largest reserves of carbon in the environment [39]. Thus, a positive association is expected between carbon and metal deposition. In the present study, linear regression analysis showed a significant and positive correlation between total carbon and Cu and Zn concentration in the surface sediment. Similarly, several studies reported a significant, strong, and positive correlation between organic matter content and Cu and Zn concentrations in the sediments [22,23].

The deposition of Cu significantly increased over time in the surface sediments of the H-02 wetland. However, the negative interaction between year and $\log_{10}TC$ means that the effect of $\log_{10}TC$ on $\log_{10}Cu$ concentration is, in fact, decreasing over time. This could be attributed to the slower rate of organic matter addition to the wetland as the H-02 wetland matures, which will potentially lead to a trough stage as hypothesized by Xu et al 2019 [30]. In an experiment that aimed at understanding the long-term ecosystem effects of riverine wetlands, it was reported that organic carbon content increased by 14% in the first 10 years of operation and then increased by only 8% in the following 5 years [40].

While both metals, Cu and Zn, showed similar increasing deposition patterns in the surface sediments over the years and seasons, regression models showed differences in how the main effects explained these patterns. Seasonal changes significantly affected Zn deposition but not Cu. Cool seasons tended to have lower Zn concentrations in the sediments as compared to the warm seasons. Harris et al 2020 found no seasonal effects on total Cu and Zn concentration in the surface sediments of the H-02 wetland [36]. This is an interesting discrepancy as the samples in that study were collected more recently. Therefore, further studies aiming at model validation are required.

Seasonal variation in metal removal and retention in the wetland system could be tied to the sulfur cycle. In the H-02 wetland system, sulfur dynamics were more prominent in warmer months where metals tended to form insoluble sulfides under anoxic conditions, while adsorption onto organic matter was more likely responsible for metal removal from the wastewater and subsequent retention in the sediments in the cooler months [30]. Harris et al 2020, found that the acid volatile sulfide (AVS) content in the surface sediments was season dependant, and was less abundant in the cooler months, which enhanced metal mobility and thus resulted in lower metal retention in the sediments in the form of metal sulfides [36]. In another study, the concentrations of Cu in the sediments of a natural March were dependent on the season, where Cu concentrations were higher in late summer to early fall, whereas Zn concentrations were the highest during summer and fall, and both metals had lowest concentrations in winter [25].

### 4.4. Modeling the deposition of metals in sediments

In this study, we used the generalized linear model to understand the effects of time and seasons on Cu and Zn deposition in the sediments. Likewise, generalized linear model helped identifiying sources of metal contamination in water and the surface sediment over time and seasons [41]. The deposition of Zn in Norwegian marine sediments over 25 years helped identifying the historical point sources responsible for metal pollution using the generalized additive models [42]. Symader and Bierl, 2000 adopted time series analysis to evaluate metal

concnetration temporal variabilities over 6 years in the sediments of a river in Germany, which eventually helped understand the sediment chemistry [43].

## 5. Conclusions

Deposition of heavy metals (Cu and Zn) in the sediment of the H-02 constructed wetland was dependent on total carbon, time, and seasons. Total carbon had a significant effect on the deposition of both metals in the surface sediment. Seasonal effects only impacted Zn deposition were cool seasons tended to accumulate less Zn. While this study focused on the prediction of the deposition patterns of total Cu and Zn in the surface sediments of the H-02 constructed wetland system, more work is needed to study the speciation and the partitioning of Cu and Zn to organic matter and clay minerals, especially in the surface sediment. Also, these studies could help understand the long-term interaction between the biogeochemical cycle of Cu and Zn and the sulfur cycle.

Heavy metal accumulation in the H-02 wetland system is an going work, and future studies should include more advanced staistical approaches including machine learning tools, which could be used to model the non-linear behavior of metal accumulation in the wetland sediments. Model validation is also necessary to assess model efficiency in predicting the depostion patterns of total metal concentrations. Future work should also include studies that focus on the effect of Cu and Zn accumulation in the surface sediments on the benthic organisms and the microbial community over time.

## Supporting information

**S1 Fig. A schematic diagram of the H-02 Wetland system.**
(DOCX)

**S2 Fig. Flowchart of the model selection process.**
(DOCX)

**S3 Fig. Relationship between log transformed total carbon ($\log_{10}$TC) and total nitrogen ($\log_{10}$TN).**
(DOCX)

**S4 Fig. Standardized residuals vs fitted values for the linear model (lm) for Cu.**
(DOCX)

**S5 Fig. Pearson's normalized residuals denoted by E for the linear model (lm) for Cu.**
(DOCX)

**S6 Fig. Pearson's normalized residuals for the linear model (lm) for Cu plotted against seasons (warm and cool), cells (1 and 2), and the years of the study (2007–2013).**
(DOCX)

**S7 Fig. Standardized residuals vs fitted values for the linear model with generalized least squares extension (gls) for Zn.**
(DOCX)

**S8 Fig. Pearson's normalized residuals for the linear model with generalized least squares extension (gls) for Zn.**
(DOCX)

**S9 Fig. Pearson's normalized residuals for the linear model with generalized least squares extension (gls) for Zn plotted against seasons (warm and cool), cells (1 and 2), and the**

**years of the study (2007–2013).**
(DOCX)

**S10 Fig. Pearson's normalized residuals for the linear model with generalized least squares extension (gls) for Zn plotted against $\log_{10}TC$ and $\log_{10}TN$.**
(DOCX)

**S1 Table. Variance inflation factors (VIF) of fixed effects included in the generalized linear model for Cu.**
(DOCX)

**S2 Table. Concentration of Cu (mg kg$^{-1}$ dry weight) in each sediment layer (Top, Middle, and Bottom) per year.**
(DOCX)

**S3 Table. Concentration of Zn (mg kg$^{-1}$ dry weight) in each sediment layer (Top, Middle, and Bottom) per year.**
(DOCX)

**S1 Text. Model selection.**
(DOCX)

**S1 File.**
(R)

**S2 File.**
(CSV)

**S3 File.**
(R)

**S1 Raw data.**
(XLSX)

## Acknowledgments

We would like to thank John Perry, Annah Nieman, Alexis Korotasz, Kara Norris, Jasmine Parks, Savannah Harris, Rebecca Philipps, Shelby Weathersbee, Cher Nicholson, Angela Lindell, Guha Dharmarajan and Raven Bier from the Savannah River Ecology Laboratory for their assistance on this study. We also want to thank Dr. Gary Mills who established this project and served as the primary investigator from 2007 to 2017.

## Author Contributions

**Conceptualization:** Xiaoyu Xu.

**Data curation:** Zeinah Elhaj Baddar.

**Formal analysis:** Zeinah Elhaj Baddar.

**Methodology:** Zeinah Elhaj Baddar.

**Project administration:** Xiaoyu Xu.

**Resources:** Xiaoyu Xu.

**Software:** Zeinah Elhaj Baddar.

**Supervision:** Xiaoyu Xu.

**Writing – original draft:** Zeinah Elhaj Baddar.

**Writing – review & editing:** Zeinah Elhaj Baddar, Erin Peck, Xiaoyu Xu.

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
