## [Decision Letter · Decision Letter 0]

11 May 2021

PONE-D-21-10299

Temporal Deposition of Copper (Cu) and Zinc (Zn) in the Sediments of Metal Removal Constructed Wetlands

PLOS ONE

Dear Dr. Elhaj Baddar,

Thank you for submitting your manuscript to PLOS ONE. After careful consideration, we feel that it has merit but does not fully meet PLOS ONE’s publication criteria as it currently stands. Therefore, we invite you to submit a revised version of the manuscript that addresses the points raised during the review process.

We look forward to receiving your revised manuscript.

Kind regards,

Zaher Mundher Yaseen

Academic Editor

PLOS ONE

Journal Requirements:

2. In your Methods section, please provide additional location information of the sampling sites, including geographic coordinates for the data set if available.

Reviewers' comments:

Reviewer's Responses to Questions

**Comments to the Author**

1. Is the manuscript technically sound, and do the data support the conclusions?

Reviewer #1: Yes

Reviewer #2: Partly

Reviewer #3: Yes

Reviewer #4: Partly

2. Has the statistical analysis been performed appropriately and rigorously? 

Reviewer #1: Yes

Reviewer #2: Yes

Reviewer #3: Yes

Reviewer #4: No

3. Have the authors made all data underlying the findings in their manuscript fully available?

Reviewer #1: Yes

Reviewer #2: Yes

Reviewer #3: Yes

Reviewer #4: Yes

4. Is the manuscript presented in an intelligible fashion and written in standard English?

Reviewer #1: Yes

Reviewer #2: Yes

Reviewer #3: Yes

Reviewer #4: No

5. Review Comments to the Author

Reviewer #1: I read the manuscript and found it very good, as it represents a scientific research of original data with an applied character that simulates the environmental treatment of heavy elements (Cu and Zn). Field experiments, statistics and chemical analyses have been performed in high technical standard and results were presented in a clear manner mostly except for some places of the manuscript. The manuscript meets all applicable standards for the ethics of experimentation and research integrity. However, I recommend to accept the manuscript after major revision and the following points should be addressed:

1. Title must be concise and informative, so the abbreviation Cu and Zn should be deleted and title become “Temporal Deposition of Copper and Zinc in the Sediments of Metal Removal Constructed Wetlands”.

2. Line 59: The expression “media/soils” doesn’t be accepted, please change it in another style.

3. Line 103-104: The author stated “Briefly, the Tritium facility discharges wastewater through source pipes into a rectangular pool where water is retained to provide hydrological control”. How much is the tritium wastewater discharged through source pipes into a rectangular basin?

4. Line 123-124: Surface standing water was carefully siphoned off from the cores avoiding disturbing the flocculent layer. To what extent surface standing water can be siphoned?

5. Line 129-130: Dried sediments were passed through a 2 mm sieve. I think after sediments be dried will need gentle grinding to be sieved easily. Line 176-177: In the model selection process, we followed the protocol mentioned in Zuur et al. 2009, (Zuur et al. 177 2009). Please, it is better is not to repeat the reference “Zuur et al. 2009” twice, try to reformulate the statement. Line 181: Please do not use do not use the active voice and the pronoun “we” as mentioned in “Therefore, we only kept log10TC in the model”.by the same way, look at Line 184; “Fourth, we refitted the model with the generalized”; Line 188; “Fifth, we specified the optimum fixed structure using the backward selection”; Lines191 and 192 “first, we evaluated the homogeneity of residuals using graphic tools and the Bartlett’s test of homogeneity. Second, we checked the normality graphically and by using the….”; and lines 194-195 “We used α at 0.05 as the significance level”.

6. Line 198: I suggest to change that title to “Deposition of Cu and Zn on the interface water-sediments”.

7. In both models, we find ambiguity in which layers the copper and zinc are deposited more? and why?

8. Please explain the role of Total carbon in the absorption process and its absorption capacity.

9. The best was to analyze the minerals and diagnose the type of clay minerals. As is known, every clay mineral has specific value of cation exchange capacity.

10. It is better to make mineralogy study for the soil sample.

11. Line 351 “Heavy metal (Cu and Zn)”, I think lead and zinc represent a plural state, not singular, so it is better to write the term heavy metals, not heavy metal.

12. If you use British language, the Acknowledgments” write as “Acknowledgements”.

Salih M Awadh

salih.awad@sc.uobaghdad.edu.iq

Reviewer #2: 1. Avoid using the first person pronouns "I or we” in your writing, and the most common reason given for this is that readers may regard such writing as being subjective, whereas science is all about objectivity.

2. manuscript should have some novelty in its work

3. remove “map” from the title of Fig S1 and line 101

4. fig S1 need to add the size and the scale

5. My suggestion to add Figure of study area to the manuscript

6. the methods section very long , please minimize it

7. the conclusion section very short and need more info, you need make balance between sections

8. Fig S5, S7, S8 and S9 titles need modify according to the charts type.

Reviewer #3: This research predicted Copper (Cu) and Zinc (Zn) deposition patterns , the research examined and studied the effects of time, seasons, and total carbon (TC) on Copper (Cu) and Zinc (Zn) deposition in the surface sediments of the H-02 constructed wetland on the Savannah River Site .

This research Covering both warm (April-September) and cool (October-March) seasons, several sediment cores were collected twice a year from the constructed wetland cells over 7 years, from 2007 to 2013.

This research used the Generalized Least Squares (GLS) & Linear Regression Model (LRM), linear regression model explained the behaviour and the variability of Cu deposition in the sediments , using the (GLS) extension with the (LRM)allowed for unequal variance .

Reviewer #4: This is an original research paper on developing a pattern evaluation of heavy metals (Cu, Zn) settlement in the surface constructed wetland by time and Total Carbon (TC). The topic selected by the authors is appreciated in the specific domain of science and engineering for several purposes like contamination treatment at a lower price and lower environmental impact concern. Nonetheless, the work needs to be improved to reach the level of publication for readers of the respected PLOS ONE journal.

Abstract:

1. This needs precise information on how the outcomes of the linear regression model are going to be impacted.

2. Cu and Zn changed the value reported so TC needs to report as well.

3. Line 40: …On the other hand, using the generalized least squares extension with the linear regression model allowed for unequal variance, and thus was more successful in explaining Zn deposition pattern… what does mean ‘more successful’? rewrite it.

Introduction:

4. Line50: … relatively cheaper… how? Explain it in detail.

5. The review assessment has been poorly drafted. Add more recent (5 years) papers.

https://link.springer.com/article/10.1007/s11356-020-11775-z

https://www.sciencedirect.com/science/article/abs/pii/S0048969721001388

6. There is no text belong why the authors used generalized generalized least squares extension with the linear regression model while there are more advanced approaches available?

Methods:

7. Need more explanation in Study site: write all the tributary or contribution to the site.

8. Why authors used limited no. of influencing parameters? Is there any problem to get more no. of those parameters metrological and/or climatological?

9. Write all functions of the code used in the ‘Italic’ font.

Results and Discussion:

10. Need statistical examination such as min, max, sd, etc in Table 1

11. FigureS1: replace with more presentable with all needed information

12. Better draw PCA Biplot to analyzed the dim strength between the factors.

13. Add regression equation in all scatter plots for showing the correlation mathematically how stronger?

14. Table S2: better add into the respective graph

15. In discussion: add how this changed value (within the used years) have an impact on the environment and local community and what measure should take to mitigate it for example several adsorption studies have been applied:

https://www.sciencedirect.com/science/article/abs/pii/S0045653521006317

https://link.springer.com/article/10.1007/s11356-021-12836-7

Conclusion:

16. Add the weakness of the study.

17. Add future objective of the research including the applying machine learning approach to predict the sediment for example

https://www.sciencedirect.com/science/article/abs/pii/S0304389420314783

https://www.sciencedirect.com/science/article/abs/pii/S026974912036351X

6. PLOS authors have the option to publish the peer review history of their article (what does this mean?). If published, this will include your full peer review and any attached files.

Reviewer #1: **Yes: **Salih Muhammad Awadh

Reviewer #2: No

Reviewer #3: No

Reviewer #4: No

---

## [Author Response · Author response to Decision Letter 0]

28 Jun 2021

Response to Reviewers

Disclaimer: Please don’t mind the inconsistencies in line numbers on the track changes-version, this is a common issue in word

Reviewer 1

I consider this paper very good and is an important addition to the literature , this research predicted Copper (Cu) and Zinc (Zn) deposition patterns , the research examined and studied the effects of time, seasons, and total carbon (TC) on Copper (Cu) and Zinc (Zn) deposition in the surface sediments of the H-02 constructed wetland on the Savannah River Site .

This research Covering both warm (April-September) and cool (October-March) seasons, several sediment cores were collected twice a year from the constructed wetland cells over 7 years, from 2007 to 2013. 

 This research used the Generalized Least Squares (GLS) & Linear Regression Model (LRM), linear regression model explained the behaviour and the variability of Cu deposition in the sediments , using the (GLS) extension with the (LRM)allowed for unequal variance .

Here are several comments :

1- Interesting and within the journal scope manuscript and it is properly organized .

2- Have appropriate forms and have academic value.

3- Methodology flowchart for the modelling procedure is needed.

We have created a flow-chart, please refer to figure S2 in the SI file

4- The data interpreted and analysed statistically in appropriate manner.

5- Presents results in a clear way .

6- The conclusions reached are properly validated by the results. 

7- The paper (Approved) ,can accepted for publication with minor revisions .

We thank the reviewer for their comments.

Reviewer 2

I read the manuscript and found it very good, as it represents a scientific research of original data with an applied character that simulates the environmental treatment of heavy elements (Cu and Zn). Field experiments, statistics and chemical analyses have been performed in high technical standard and results were presented in a clear manner mostly except for some places of the manuscript. The manuscript meets all applicable standards for the ethics of experimentation and research integrity. However, I recommend to accept the manuscript after major revision and the following points should be addressed:

1. Title must be concise and informative, so the abbreviation Cu and Zn should be deleted and title become “Temporal Deposition of Copper and Zinc in the Sediments of Metal Removal Constructed Wetlands”.

We appreciate the reviewer’s comment. However, keeping the elemental formula for both metals improves the searchability and visibility of our article to other researchers.

2. Line 59: The expression “media/soils” doesn’t be accepted, please change it in another style.

We fixed this issue, please refer to line 67.

3. Line 103-104: The author stated “Briefly, the Tritium facility discharges wastewater through source pipes into a rectangular pool where water is retained to provide hydrological control”. How much is the tritium wastewater discharged through source pipes into a rectangular basin?

We thank the reviewer for this comment. We don’t have access to such data.

4. Line 123-124: Surface standing water was carefully siphoned off from the cores avoiding disturbing the flocculent layer. To what extent surface standing water can be siphoned?

Ideally, we were careful not to disturb/siphon the flocculant layer (soft organic rich floccule-like material). The amount of surface standing water in sediment core was variable depending on the water level in the wetland at the time of sampling.

5. Line 129-130: Dried sediments were passed through a 2 mm sieve. I think after sediments be dried will need gentle grinding to be sieved easily. The sediments were only sieved, no grinding was applied. Line 176-177: In the model selection process, we followed the protocol mentioned in Zuur et al. 2009, (Zuur et al. 177 2009). Please, it is better is not to repeat the reference “Zuur et al. 2009” twice, try to reformulate the statement. Line 181: Please do not use do not use the active voice and the pronoun “we” as mentioned in “Therefore, we only kept log10TC in the model”.by the same way, look at Line 184; “Fourth, we refitted the model with the generalized”; Line 188; “Fifth, we specified the optimum fixed structure using the backward selection”; Lines191 and 192 “first, we evaluated the homogeneity of residuals using graphic tools and the Bartlett’s test of homogeneity. Second, we checked the normality graphically and by using the….”; and lines 194-195 “We used α at 0.05 as the significance level”. 

We have fixed all these issues, please refer to lines 192-197 in the manuscript and Text S1 in the SI file.

6. Line 198: I suggest to change that title to “Deposition of Cu and Zn on the interface water-sediments”. 

We appreciate the reviewer comment, but we analysed the metals in the whole sediment cores and not just the very top part in contact with the standing water above. 

7. In both models, we find ambiguity in which layers the copper and zinc are deposited more? and why? 

Only surface sediments metal data were used in the model. We have mentioned the rationale behind this in the text (Lines 176-179), but briefly, the surface sediments have considerably higher Cu, Zn, and carbon content than the layers underneath. 

8. Please explain the role of Total carbon in the absorption process and its absorption capacity.

We have not performed adsorption isotherm experiments that would help figure out the adsorption process or capacity. While this would be an interesting aspect, it was beyond the scope of the current study.

9. The best was to analyze the minerals and diagnose the type of clay minerals. As is known, every clay mineral has specific value of cation exchange capacity.

We thank and agree with the reviewer. However, we have not performed such analysis.

10. It is better to make mineralogy study for the soil sample.

We appreciate the reviewer comment, this would be an interesting analysis for another follow up study. In this work, we were particularly interested in heavy metal accumulation trend over the years and seasons.

11. Line 351 “Heavy metal (Cu and Zn)”, I think lead and zinc represent a plural state, not singular, so it is better to write the term heavy metals, not heavy metal.

We appreciate the reviewer comment, please refer to line 540.

12. If you use British language, the Acknowledgments” write as “Acknowledgements”.

We appreciate the reviewer comment, please refer to line 556.

Reviewer 3

Overall, the calibration and validation processes are acceptable. So, I found the manuscript in a favorable format to be published on PLOS ONE journal with the topic “Temporal Deposition of Copper (Cu) and Zinc (Zn) in the Sediments of Metal Removal Constructed Wetlands”. This research project has the scientific background enough to be published, and the authors present a fair amount of data and a reasonable list of references. However, there is some comments needed to be observing. 

1. Avoid using the first person pronouns "I or we” in your writing, and the most common reason given for this is that readers may regard such writing as being subjective, whereas science is all about objectivity.

We have fixed all these issues, please refer to lines 192-197 in the manuscript and Text S1 in the SI file.

2. manuscript should have some novelty in its work

We thank the reviewer for the feedback. We think that this works novelty comes from our use of statistical tools and 7 years- worth of data to give us insights about the accumulation pattern of heavy metals in the surface sediments of the H-02 wetland. We believe that this is an invaluable tool which we could build upon for the next years to predict the performance and thus the efficiency of the H-02 wetland, and to protect the environment and the wildlife at the Savannah River site.

3. remove “map” from the title of Fig S1 and line 101

We have replaced the word “map” with: Schematic diagram”, line 112, please refer to Fig S1 title (line 570), and Fig S1 title in the SI Paper-Track changes file.

4. fig S1 need to add the size and the scale 

We have updated the figure with the required information

5. My suggestion to add Figure of study area to the manuscript

We have already provided a schematic diagram of the study area (please see Fig S1 in the SI file). Also, we don’t have access to an airborne image of the study area.

6. the methods section very long , please minimize it 

We fixed this issue by moving model selection details to the SI file, please refer to Text S1 in the SI file and lines 192-197 in the manuscript.

7. the conclusion section very short and need more info, you need make balance between sections

We fixed this issue, please refer to lines 540-554. 

8. Fig S5, S7, S8 and S9 titles need modify according to the charts type.

We have modified the titles, please notice that the figure numbers have been updated, please see Fig S6, S8, S9, and S10.

Reviewer 4

This is an original research paper on developing a pattern evaluation of heavy metals (Cu, Zn) 

settlement in the surface constructed wetland by time and Total Carbon (TC). The topic 

selected by the authors is appreciated in the specific domain of science and engineering for 

several purposes like contamination treatment at a lower price and lower environmental impact 

concern. Nonetheless, the work needs to be improved to reach the level of publication for 

readers of the respected PLOS ONE journal. 

Abstract: 

1. This needs precise information on how the outcomes of the linear regression model are 

going to be impacted. 

We believe that we have provided precise and summarized details of the linear regression model in the abstract (lines 40-45), we would appreciate it if the reviewer would give us more clarification of what exactly we should be adding in the abstract. 

2. Cu and Zn changed the value reported so TC needs to report as well. 

We thank the reviewer for this comment. But since we were more focused on Cu and Zn accumulation, we dropped the TC from line 37.

3. Line 40: …On the other hand, using the generalized least squares extension with the 

linear regression model allowed for unequal variance, and thus was more successful in 

explaining Zn deposition pattern… what does mean ‘more successful’? rewrite it. 

We added a clarifying sentence, please refer to lines 41-43.

Introduction: 

4. Line50: … relatively cheaper… how? Explain it in detail.

We added a sentence to clarify our point, please refer to lines 50-51.

5. The review assessment has been poorly drafted. Add more recent (5 years) papers. 

We added more recent references, please check lines 52, 54, 68-69, 77, 81, 82, 84

https://link.springer.com/article/10.1007/s11356-020-11775-z

https://www.sciencedirect.com/science/article/abs/pii/S0048969721001388

6. There is no text belong why the authors used generalized generalized least squares 

extension with the linear regression model while there are more advanced approaches 

available? 

The generalized least squares (GLS) extension is used to handle residual heterogeneity. We added this clarification in line 115.

Methods: 

7. Need more explanation in Study site: write all the tributary or contribution to the site. 

There is no tributary or contribution to the site

8. Why authors used limited no. of influencing parameters? Is there any problem to get 

more no. of those parameters metrological and/or climatological? 

We thank the reviewer for their comment. We have not measured these parameters.

9. Write all functions of the code used in the ‘Italic’ font. 

We have made the required changes. Please refer to Text S1 in the SI file.

Results and Discussion: 

10. Need statistical examination such as min, max, sd, etc in Table 1 

We have added the maximum, minimum, and replaced 95% confidence intervals with the standard deviation. Please refer to table 1 in the manuscript as well as lines 210 and 214.

11. FigureS1: replace with more presentable with all needed information 

We have added the required information, please see Fig. S1 in the SI file.

12. Better draw PCA Biplot to analyzed the dim strength between the factors.

We appreciate the reviewer comment. However, in this work, we were more interested in addressing the accumulation behaviour of the metals in the wetland surface sediment over time and seasons. We will definitely consider PCA as a valuable tool in our upcoming work.

13. Add regression equation in all scatter plots for showing the correlation mathematically 

how stronger? 

We removed table S2 and added regression equations and correlation coefficients to the scatter plots. Please refer to “Fig3NEW” in the attachments.

14.Table S2: better add into the respective graph 

Please see our response to comment 13 above.

15. In discussion: add how this changed value (within the used years) have an impact on 

the environment and local community and what measure should take to mitigate it for 

example several adsorption studies have been applied: 

https://www.sciencedirect.com/science/article/abs/pii/S0045653521006317

https://link.springer.com/article/10.1007/s11356-021-12836-7

We thank the reviewer for this comment, but we have not performed any studies that would answer this question.

Conclusion: 

16. Add the weakness of the study. 

We have added weaknesses and future directions in the conclusion section (lines 548-562)

17. Add future objective of the research including the applying machine learning approach to predict the sediment for example 

https://www.sciencedirect.com/science/article/abs/pii/S0304389420314783

https://www.sciencedirect.com/science/article/abs/pii/S026974912036351X

We have added a sentence that addressed this suggestion, please refer to lines 557-559

---

## [Decision Letter · Decision Letter 1]

19 Jul 2021

Temporal Deposition of Copper (Cu) and Zinc (Zn) in the Sediments of Metal Removal Constructed Wetlands

PONE-D-21-10299R1

Dear Dr. Elhaj Baddar,

We’re pleased to inform you that your manuscript has been judged scientifically suitable for publication and will be formally accepted for publication once it meets all outstanding technical requirements.

Kind regards,

Zaher Mundher Yaseen

Academic Editor

PLOS ONE

Additional Editor Comments (optional):

Reviewers' comments:

Reviewer's Responses to Questions

**Comments to the Author**

1. If the authors have adequately addressed your comments raised in a previous round of review and you feel that this manuscript is now acceptable for publication, you may indicate that here to bypass the “Comments to the Author” section, enter your conflict of interest statement in the “Confidential to Editor” section, and submit your "Accept" recommendation.

Reviewer #1: All comments have been addressed

Reviewer #3: (No Response)

2. Is the manuscript technically sound, and do the data support the conclusions?

Reviewer #1: Yes

Reviewer #3: (No Response)

3. Has the statistical analysis been performed appropriately and rigorously? 

Reviewer #1: Yes

Reviewer #3: (No Response)

4. Have the authors made all data underlying the findings in their manuscript fully available?

Reviewer #1: Yes

Reviewer #3: (No Response)

5. Is the manuscript presented in an intelligible fashion and written in standard English?

Reviewer #1: Yes

Reviewer #3: (No Response)

6. Review Comments to the Author

Reviewer #1: I have reviewed the required amendments and found a good response from the researcher. He accomplished most of the basic points and answered some of them adequately. I therefore find the manuscript now is more quality, so I recommend to give an accept decision.

Additionally, I advise the author that I prefer to remove the chemical element symbols (Cu and Zn) from the manuscript title. It doesn’t mean thing to be included in the title.

Reviewer #3: (No Response)

7. PLOS authors have the option to publish the peer review history of their article (what does this mean?). If published, this will include your full peer review and any attached files.

Reviewer #1: **Yes: **Salih Muhammad Awadh

Reviewer #3: No

---

## [Editor Report · Acceptance letter]

26 Jul 2021

PONE-D-21-10299R1 

Temporal deposition of copper and zinc in the sediments of metal removal constructed wetlands 

Dear Dr. Elhaj Baddar:

I'm pleased to inform you that your manuscript has been deemed suitable for publication in PLOS ONE. Congratulations! Your manuscript is now with our production department. 

Kind regards, 

on behalf of

Dr. Zaher Mundher Yaseen 

Academic Editor

PLOS ONE